# Self-Prompt: Leveraging Gradient-Based Search for Optimizing Prompts in Code Generation

## Abstract

With the rapid advancement of large language models, code generation has witnessed substantial progress. As a crucial factor influencing the quality of generated code, prompt design plays a pivotal role in optimizing model performance. While manual prompt engineering remains a common approach, it is often labor-intensive and suboptimal. To address these limitations, automated prompt optimization techniques have been introduced. However, existing methods that rely on LLMs for automatic prompt construction are inherently constrained by the models' own capabilities, leading to inconsistencies in code generation quality. In this paper, we propose Self-Prompt, an automated prompt engineering framework tailored for code generation tasks, designed to enhance code quality while ensuring stability and progressive refinement. By leveraging task-specific data, Self-Prompt formulates prompt optimization as a search problem, effectively transforming LLM-based code generation into an iterative prompt refinement process. To assess the effectiveness of Self-Prompt, we conduct extensive experiments using five open-source LLMs across three widely adopted code generation benchmarks: HumanEval, MBPP, and EvalPlus. Employing pass@k as the primary evaluation metric, our results demonstrate that Self-Prompt achieves performance comparable to or exceeding state-of-the-art prompt engineering methods, highlighting its potential for improving automated code generation.

## 1 Introduction

With the continuous advancement of Large Language Models (LLMs), their ability to generate accurate and functional code has significantly improved, contributing to enhanced human productivity Zan et al. (2023a); Chen et al. (2024). Despite these advancements, LLMs still face inherent limitations in code generation, frequently producing outputs that contain errors or exhibit inconsistencies, thereby affecting overall reliability Zhang et al. (2023). To address the challenges in code generation and improve performance, existing strategies predominantly fall into two categories: fine-tuning and prompt engineering. Fine-tuning approaches (illustrated in Figure 2 (a)), as discussed in Hui et al. (2024); Yu et al. (2024); Ding et al. (2024); Wang et al. (2024b), generally involve adapting a base LLM by training it on coding datasets. While effective, these methods require significant hardware resources, specialized training techniques, and high-quality coding datasets. Additionally, ensuring the model's generalization ability post-finetuning requires extensive validation, often necessitating multiple trials to guarantee stable and reliable performance on unseen data. On the other hand, prompt engineering (illustrated in Figure 2 (b)) offers an alternative method for improving code generation accuracy. Although the inherent limitations of LLMs establish a ceiling for code generation performance, the design of prompts—serving as the critical interface between humans and LLMs—plays a crucial role in achieving this potential. While manual prompt engineering is often the preferred approach due to its direct controllability, constructing high-quality prompts remains a challenging task. It typically involves numerous interactions with the LLMs and iterative refinements based on feedback, a process that demands substantial time and empirical expertise. Existing automated prompt engineering approaches, such as Automatic Prompt Engineer (APE) Zhou et al. (2023), attempt to tackle the challenges of prompt design by utilizing LLMs as self-sufficient prompt engineers. APE generates candidate prompts, followed by quality evaluation, demonstrating that

machine-generated prompts can either match or exceed the performance of manually crafted ones. However, in the context of code generation, APE faces two critical limitations: 1) its performance is heavily reliant on the base LLM's prompt construction capabilities—insufficient capabilities can lead to suboptimal prompts that contain overly specific coding biases (as illustrated in Figure 1 with ChatGLM3-6B GLM et al. (2024) on the HumanEval Chen et al. (2021) dataset), sometimes resulting in poorer performance than default prompts; 2) the framework lacks robust mechanisms for post-selection prompt refinement.

To address these challenges, we propose Self-Prompt, an automated prompt generation framework specifically designed for code generation tasks. Our framework adopts a two-phase architecture:

1. Base Prompt Generation: Expanding on the foundation established by APE, we utilize LLMs to generate the initial prompts while incorporating a prompt filter designed to eliminate prompts that introduce significant coding biases.

2. Extra Prompt Generation: We introduce an innovative, training-aware mechanism that identifies code-sensitive tokens, which, when integrated into the prompts, enhance the quality of generated code.

Our key contributions are as follows:

- We introduce a novel methodology for identifying code-sensitive tokens during model training, facilitating a systematic approach to enhancing code generation quality.
- We propose Self-Prompt, an automated prompt engineering framework that leverages the intrinsic capabilities of LLMs to generate and optimize prompts dynamically.
- We conduct extensive empirical evaluations on three widely used code generation benchmarks (HumanEval, MBPP, and EvalPlus) using five state-of-the-art open-source LLMs. Our results demonstrate that Self-Prompt achieves competitive performance, surpassing existing prompt engineering techniques in effectiveness and stability.

| Method | Prompt Content | Score |
|---|---|---|
| General | You are a helpful assistant. | 0.366 |
| LLM-generated Prompt (APE) | The instruction was to write two functions, 'sum_product' and 'rolling_max', based on the given inputs. The 'sum_product' function takes a list of integers as input and returns a tuple consisting of the sum and the product of all the integers in the list. If the list is empty, it should return (0, 1). The 'rolling_max' function takes a list of integers as input and returns a list of rolling maximum elements found until a given moment in the sequence. If the input list is empty, it should return an empty list. Please provide the two functions you have written based on the given inputs. | 0.262 |

Figure 1: A real-world case study of APE using ChatGLM-6B as the backbone model and HumanEval as the benchmark dataset for evaluation. The prompts generated by the LLM exhibit a noticeable bias, ultimately resulting in degraded performance.

## 2 RELATED WORK

### 2.1 CODE GENERATION

Code generation denotes the systematic translation process that converts natural language specifications of programming requirements into functionally equivalent executable code Li et al. (2023);

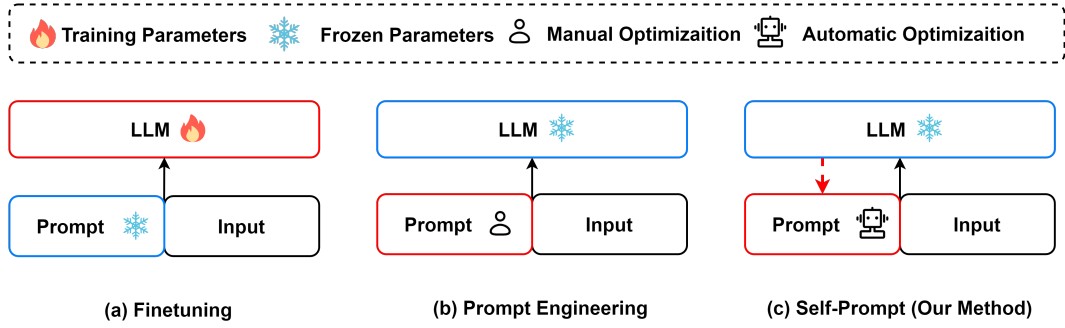

Figure 2: Comparison of the paradigms of fine-tuning, prompt engineering, and our proposed method, Self-Prompt. The key difference between prompt engineering and our approach lies in the utilization of embedding layer training within LLMs to enable automatic prompt optimization. In (c), the red dotted line indicates the process of searching for an optimized prompt within the model itself, representing one of our key contributions.

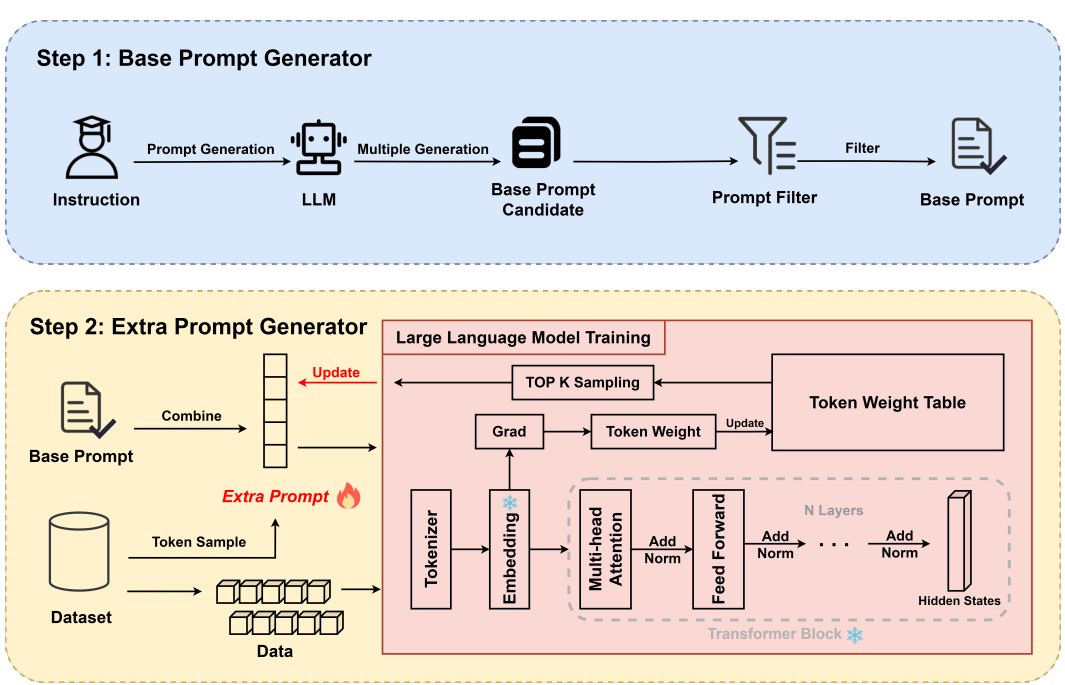

Figure 3: The workflow of Self-Prompt, demonstrated through a real-world example. The process begins with the Base Prompt Generator, which produces an initial code-related prompt. Subsequently, the Extra Prompt Generator refines the prompt through an optimization process.

Zan et al. (2023b). Within this research domain, the primary focus lies in generating code that meets functional correctness criteria. Current methodologies predominantly involve fine-tuning specialized pre-trained models, including but not limited to Qwen 2.5 Coder Hui et al. (2024), Yi Coder 01.AI (2024), CodeGeeX 4 Zheng et al. (2023), DeepSeek Coder V2 Zhu et al. (2024), and CodeGemma Zhao et al. (2024). Complementary approaches incorporate advanced prompt engineering techniques and multi-agent collaborative systems, as documented in Han et al. (2024); Zhang et al. (2024). Reinforcement learning paradigms have also demonstrated effectiveness in optimizing code generation processes, with notable implementations detailed in Dou et al. (2024); Liu et al. (2023a). Specialized advancements in the field include stylistic adaptation mechanisms, where Dai et al. (2024) proposes pattern alignment techniques to match user-specific coding styles.

---

**Algorithm 1** Base Prompt Generator Algorithm

---

**Input**: Initial prompt template $P_T$, Dataset $D$, Model $f$, model weights $\theta$, sample function $S$, heuristic rule $H$

1: $D_k = S(D, k)$ {Sample from Dataset $D$.}
2: $P_{BASE} = f((P_T, D), \theta)$
3: $P_{BASE} = H(P_{BASE})$
4: **return** $P_{BASE}$

---

Cross-lingual interoperability solutions are being developed concurrently, as evidenced by Paul et al. (2024)'s innovative use of intermediate representations to mitigate programming language discrepancies.

In our method, we try to realize automated code generation through prompting with LLMs, while obtaining code generation prompt with minimal cost and generating code with high quality.

## 2.2 PROMPT ENGINEERING

As a critical interface between humans and LLMs, prompts have been empirically validated as an effective mechanism for task execution, with methodologically constructed prompts demonstrating performance parity with supervised fine-tuning approaches Li et al. (2024); Wang et al. (2024a). Notwithstanding their demonstrated efficacy, the development of sophisticated prompting strategies presents non-trivial challenges, primarily attributed to the lexical sensitivity of prompt components Yang et al. (2024b) and the consequent necessity for domain expertise in prompt engineering. The discipline of Prompt Engineering has consequently emerged as a formal research domain, seeking to establish systematic methodologies (both manual and algorithmic) for optimized prompt formulation. Current research findings indicate that strategic integration of task-specific examples within prompts can significantly enhance model performance through in-context learning Brown et al. (2020). A seminal advancement in this field is APE Zhou et al. (2023), which pioneers the concept of meta-prompting by enabling LLMs to autonomously generate and refine task-appropriate prompts through an iterative refinement process. The APE framework operates through three core phases: (1) LLM-based prompt generation, (2) multi-dimensional performance evaluation using task-specific metrics, and (3) evolutionary optimization of prompt candidates. In this paper, we employed APE as our baseline comparison method while utilizing general-purpose LLMs rather than specifically code-fine-tuned variants for our experimental framework. This methodological selection was driven by our approach's dual requirement for both prompt engineering capabilities and code generation competencies. General-purpose LLMs have superior balance between these two critical dimensions compared to their specialized counterparts, thereby ensuring optimal performance equilibrium in our code generation task execution.

## 3 METHODOLOGY

### 3.1 PROBLEM DEFINITION

We present a rigorous mathematical formulation of Self-Prompt's optimization framework. Let $D$ denote the input space of natural language code descriptions and $C$ represent the output code space. The code generation task can be formally defined as establishing a mapping function $F : D \rightarrow C$. To characterize the learning dynamics, we define $M$ as the semantic distribution space of model parameters and $L$ as the target distribution space of ground truth labels. The training procedure constitutes a distribution transformation $\Delta T : M(D) \rightarrow C \leftrightarrow L$ , where $\Delta T$ encapsulates the distributional shift during learning.

The Kullback-Leibler divergence provides a quantitative measure for this distribution transformation:

$$D_{\text{KL}}(\mathcal{M}(\mathcal{D}) \parallel \mathcal{L}) \tag{1}$$

The optimization objective involves minimizing this divergence between the model's distribution $M(D)$ and the target distribution $L$:

$$\mathcal{L}\text{obj} = \arg\min \theta \big( D_{\text{KL}}(\mathcal{M}(\mathcal{D}) \parallel \mathcal{L}) \big) \tag{2}$$

---

**Algorithm 2** Extra Prompt Generator Algorithm

---

**Input**: Dataset $D$, Model $f$, model weights $\theta$, vocabulary embedding $E^{|V|}$, optimization steps $T$, learning rate $\gamma$, sample function $S$, weight coefficient function $\alpha$

1: $P_{EXTRA} = S(D_{INPUT}, m)$ {Sample m tokens from Dataset input.}
2: $P = P_{BASE} + P_{EXTRA}$
3: **for** $1, ..., T$ **do**
4:  Retrieve mini-batch $(X, Y) \subseteq D$.
5:  $L_{TASK} = f((P, X_i), Y_i, \theta)$ {Calculating loss on given task.}
6:  $g = \nabla_{E^{|V|}} L_{TASK}$ {Calculating gradients on vocabulary embedding.}
7:  $W = \alpha W$ {Update weight table.}
8:  $P^*_{EXTRA} = TOPN(W \odot g)$ {Select the top n tokens with the highest gradients.}
9:  $E^{|V|} = E^{|V|} - \gamma g$
10: **end for**
11: **return** $P^*_{EXTRA}$

---

where $L_{obj}$ denotes the objective function and $\theta$ represents the model parameters. The Self-Prompt optimization mechanism $f$ adaptively captures the distributional shift from source to target domains, formally expressed as:

$$f \sim D_{\text{KL}}(\mathcal{M}(\mathcal{D}) \parallel \mathcal{L}) \tag{3}$$

## 3.2 SELF-PROMPT

### 3.2.1 BASE PROMPT GENERATOR

The workflow of the Base Prompt Generator is formally defined in Algorithm 1. To facilitate automated generation of task-specific prompts, we first construct a structured prompt template $P_T$ designed to guide the LLMs' generation process. To enhance alignment between the generated prompts and task requirements, we implement a systematic sampling approach by selecting $k$ distinct input-output pairs from the target task dataset through simple random sampling without replacement. This combinatorial sampling strategy can be mathematically represented as:

$$\binom{n}{k} = \frac{n!}{k!(n-k)!} \tag{4}$$

where $n$ denotes the total number of available instances. The concatenated input comprising $P_T$ and the sampled demonstrations is then processed by LLMs to generate candidate prompts. Subsequently, we apply a set of heuristic filtering rules to eliminate prompts demonstrating overt coding tendencies or syntactic irregularities. The refined outputs are subsequently aggregated to construct the final Base Prompt $P_{BASE}$, ensuring both linguistic quality and functional relevance to the target task.

### 3.2.2 EXTRA PROMPT GENERATOR

To refine the prompts generated by the Base Prompt Generator for better performance on given tasks, we require an optimization process. Hence, we propose the Extra Prompt Generator, which operates under our novel framework for prompt optimization, advancing the performance of base prompts. Throughout the creation of the Extra Prompt, we explore the interplay between gradients and loss, leading to the delineation of useful and harmful prompts contingent upon the loss scenario.

The process of Extra Prompt Generator is concretely defined in Algorithm 2. Firstly, we sample m tokens from dataset input to generate initial Extra Prompt $P_{EXTRA}$, and combine $P_{BASE}$ with $P_{EXTRA}$ for training, $P = P_{BASE} + P_{EXTRA}$. Then we follow the conventional training process to train the model's embedding layer $E^{|V| \times d}$ (we do not alter the weight after training), where $|V|$ is the vocabulary size of the model, and $d$ is the dimension of the embedding, that is, keeping $P$ fixed, and training $E^{|V| \times d}$ based on dataset. Additionally, we use an objective function $L_{TASK}$ to represent the optimization on the given task.

Guided by Table 1 and 2, during each step, we maintain a weight table of each token $W^{|V|}$, the weight of each token will be updated according to loss at each step, through which we can separate

| Gradients | Prompt |
|-----------|--------|
| Increase | Gradients increase, which means models dislike these prompts, and these prompts are harmful. |
| Decrease | Gradients decrease, which means models prefer these prompts, and these prompts are useful. |

Table 1: The relationship between gradients and prompts.

| Definition | Explanation |
|------------|-------------|
| Useful Prompt | Be able to decrease loss, and can lead to better performance. |
| Harmful Prompt | Be able to increase loss, and can lead to worse performance. |
| Useless Prompt | Loss fluctuates slightly, and performance is not obviously affected. |

Table 2: The difference of different prompts.

useful prompts and less harmful prompts gradually, useful prompts has a higher probability to be selected and harmful has a lower probability. Assisted with loss from previous step $L_{pre}$ and current step $L_{cur}$, here we define a weight coefficient calculation function as

$$\alpha = 1 - \frac{(L_{cur} - L_{pre})}{max(L_{cur}, L_{pre})} \tag{5}$$

and the weight table will be updated

$$W \sim [w_1, w_2, ..., w_{|V|}]^\top \tag{6}$$

$$W_i = \alpha W_i, i = 1, ..., |V| \tag{7}$$

then we combine weight table with gradients $g^{|V| \times d}$, then we select m tokens with the largest gradients, which we designate as our optimization tokens.

$$P^*_{EXTRA} = TOPN(W \odot g, N = m) \tag{8}$$

Ultimately, our final optimized prompt will be

$$P_{FINAL} = P_{BASE} + P^*_{EXTRA} \tag{9}$$

## 4 EXPERIMENTS

### 4.1 EXPERIMENTAL SETTINGS

#### 4.1.1 IMPLEMENTATION DETAILS

The experimental validation was conducted on an L40 GPU platform. During the training phase, we exclusively optimized the embedding module of the model while intentionally omitting weight preservation upon training completion. The model underwent 20 training epochs using the AdamW optimizer Loshchilov & Hutter (2019), configured with a learning rate of 1e-4 and a batch size of 16. To ensure statistical reliability, all experimental results were calculated as mean values obtained from five independent trials with distinct random seeds during data generation.

#### 4.1.2 DATASETS

For the comprehensive evaluation of the proposed methodology in the NL2Code domain, we conduct systematic experiments on three benchmark datasets with distinct characteristics:

1. **HumanEval (Hand-Crafted Evaluation Set)** Chen et al. (2021): This carefully curated code generation benchmark comprises 164 hand-authored programming problems. Each

|  | ChatGLM3-6B | Qwen2-7B | Llama3-8B | Gemma2-9B | Yi-1.5-9B |
|---|---|---|---|---|---|
| General | 0.366 | 0.756 | 0.567 | 0.616 | 0.591 |
| Human | 0.488 | 0.744 | 0.573 | 0.640 | 0.604 |
| APE | 0.262 | 0.774 | 0.573 | 0.646 | 0.616 |
| Self-Prompt | **0.518** | **0.780** | **0.610** | **0.665** | **0.646** |

Table 3: The experimental results on the HumanEval dataset, using pass@1 as the evaluation metric. *General* refers to using a general prompt to generate answers. *Human* means using human-curated prompt to generate answers.

|  | ChatGLM3-6B | Qwen2-7B | Llama3-8B | Gemma2-9B | Yi-1.5-9B |
|---|---|---|---|---|---|
| General | 0.524 | 0.521 | 0.679 | 0.667 | 0.732 |
| Human | 0.598 | 0.582 | **0.698** | 0.712 | 0.738 |
| APE | 0.646 | 0.632 | **0.698** | 0.690 | **0.762** |
| Self-Prompt | **0.682** | **0.647** | 0.687 | **0.724** | 0.749 |

Table 4: The experimental results on MBPP dataset, using pass@1 as the evaluation metric.

|  | ChatGLM3-6B | Qwen2-7B | Llama3-8B | Gemma2-9B | Yi-1.5-9B |
|---|---|---|---|---|---|
| General | 0.323 | 0.726 | 0.524 | 0.579 | 0.543 |
| Human | **0.451** | 0.689 | 0.518 | 0.585 | 0.543 |
| APE | 0.232 | 0.713 | 0.518 | 0.598 | 0.561 |
| Self-Prompt | **0.451** | **0.762** | **0.567** | **0.610** | **0.585** |

Table 5: The experimental results on EvalPlus (HumanEval+) dataset, using pass@1 as the evaluation metric.

|  | ChatGLM3-6B | Qwen2-7B | Llama3-8B | Gemma2-9B | Yi-1.5-9B |
|---|---|---|---|---|---|
| General | 0.451 | 0.449 | 0.556 | 0.569 | 0.612 |
| Human | 0.495 | 0.500 | **0.585** | 0.590 | 0.608 |
| APE | 0.542 | **0.558** | **0.585** | 0.590 | 0.635 |
| Self-Prompt | **0.586** | 0.536 | 0.574 | **0.602** | **0.647** |

Table 6: The experimental results on EvalPlus (MBPP+) dataset, using pass@1 as the evaluation metric.

    sample contains: (a) a well-defined function signature, (b) a natural language specification detailing functional requirements, (c) an incomplete function implementation skeleton, and (d) multiple unit test cases (with an average of 7.7 assertions per problem). The manual construction process ensures high-quality problem formulation and test coverage.

2. **MBPP (Mostly Basic Python Problems Dataset)** Austin et al. (2021): This crowd-sourced dataset contains approximately 1,000 entry-level Python programming challenges designed for novice developers. Each instance includes: (a) a task description specifying input-output behavior, (b) a reference implementation demonstrating basic programming concepts and standard library usage, and (c) three pre-defined test cases. The collection emphasizes fundamental algorithmic patterns and practical programming constructs.

3. **EvalPlus Framework** Liu et al. (2023b): As an enhanced evaluation paradigm for program synthesis, EvalPlus introduces a systematic methodology for test case augmentation through automated input generation. The framework addresses deficiencies in existing

benchmarks by: (a) generating comprehensive test suites to detect subtle implementation errors, (b) identifying coverage gaps that may lead to inflated performance estimates, and (c) creating extended versions of popular benchmarks (HumanEval+ and MBPP+) with rigorous test case validation. Empirical validation demonstrates its effectiveness in reducing Type II evaluation errors through multi-aspect test case analysis.

### 4.1.3 EVALUATION METRICS

**pass@k Metric:** Adhering to the evaluation protocol established in the HumanEval benchmark, we employ the pass@k metric for assessing code generation performance. For each programming problem, we generate k candidate solutions for evaluation. A problem is considered successfully solved if at least one solution passes all ground-truth test cases. The pass@k score is subsequently calculated as the proportion of solved problems within the entire dataset. Consistent with mainstream evaluation practices in code generation research, we adopt k=1 as our primary evaluation configuration to align with conventional measurement standards in this field. This metric rigorously evaluates the model's ability to produce correct solutions on the first attempt, reflecting practical deployment requirements for automated code generation systems.

### 4.1.4 MODELS

To ensure methodological rigor in our comparative analysis, we deliberately selected contemporarily available open-source language models with documented strong performance across multiple benchmarks. Our curated selection comprises: ChatGLM3-6B GLM et al. (2024), Qwen2-7B Yang et al. (2024a), Llama3-8B AI@Meta (2024), Gemma2-9B Team (2024), and Yi-1.5-9B Young et al. (2024). This selection criteria specifically excluded proprietary/closed-source systems to maintain reproducibility and comparability in our benchmark experiments.

### 4.1.5 BASELINE SETTINGS

In our experimental framework, we systematically evaluate four distinct prompting configurations for LLMs: *General*, *Human*, *APE*, and *Self-Prompt*. The *General* configuration represents the default setting employed by most mainstream LLMs, utilizing the fundamental prompt "You are a helpful assistant." The *Human* configuration constitutes our primary comparative benchmark, featuring meticulously crafted manual prompts specifically optimized for code generation tasks.

## 4.2 EXPERIMENT RESULTS

The experimental results presented in Table 3, Table 4, Table 5, and Table 6 systematically demonstrate the performance superiority of our approach across various models on three benchmark datasets: HumanEval, MBPP, and EvalPlus, using pass@1 as the primary evaluation metric. Notably, our method exhibits competitive advantages compared to both human-curated prompts and APE approaches. This performance improvement can be primarily attributed to two synergistic components in our framework: (1) The Base Prompt Generator specializes in creating task-specific prompts that enhance LLMs' contextual understanding, thereby facilitating the generation of more accurate and task-aligned responses; and (2) The Extra Prompt Generator produces optimized lexical supplements that strategically increase the probability of generating crucial answer-related tokens. Through the coordinated operation of these components, our methodology achieves automated generation of high-precision prompts that simultaneously incorporate domain-specific instructions and answer-critical keywords.

## 4.3 ABLATION STUDY

### 4.3.1 MODULE OF SELF-PROMPT

To systematically evaluate the constituent components of our Self-Prompt framework, we conducted ablation studies on HumanEval using two distinct LLMs: ChatGLM3-6B and Gemma2-9B. Through quantitative analysis of experimental results presented in Table 7, we empirically validate the functional necessity and integral role of each modular component within our methodological framework.

|  | ChatGLM3-6B | Gemma2-9B |
|---|---|---|
| APE | 0.262 | 0.646 |
| Self-Prompt | **0.518** | **0.665** |
| w/o Extra Prompt | 0.476 | 0.646 |
| w/o Base Prompt | 0.341 | 0.640 |

Table 7: The different contribution of module of Self-Prompt. Experiments on HumanEval with ChatGLM3-6B and Gemma2-9B.

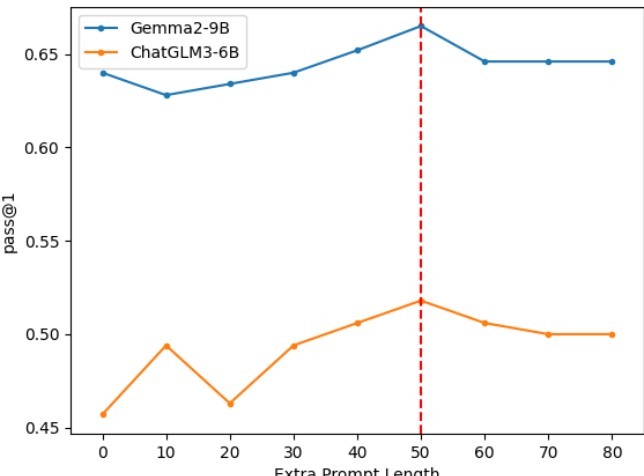

Figure 4: Ablation study on Extra Prompt Length, which we validate on the HumanEval dataset using ChatGLM3-6B and Gemma2-9B, show pass@1 scores for different prompt length settings.

Our controlled experiments specifically investigate the individual contributions of different architectural elements to the overall performance enhancement.

### 4.3.2   EXTRA PROMPT LENGTH

We proceed to conduct an ablation analysis regarding the optimal length of extra prompt tokens. As depicted in Figure 4, we validate this assertion through the utilization of both ChatGLM3-6B and Gemma2-9B on the HumanEval dataset. Our observations reveal that the exclusion of an extra prompt results in a model performance that is relatively suboptimal; conversely, the employment of an extra prompt that is excessively long fails to achieve the peak performance. This phenomenon is attributed to the inherent capacity of our method to introduce a few deleterious prompt tokens, with longer prompts potentially introducing an augmented number of such tokens, thereby exacerbating performance degradation. Our empirical findings demonstrate that the optimal performance is achieved when an extra prompt with a token length of 50 is employed.

## 5   CONCLUSION

In this study, we propose a gradient-based approach that bridges the transition from model training processes to prompt search optimization. We further present the Self-Prompt framework, which enables automatic code generation and iterative refinement of prompts for target models. Through comprehensive empirical validation in code generation tasks, we demonstrate the framework's effectiveness in prompt optimization, highlighting its capability to progressively enhance automatically generated prompts through systematic refinement processes.

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
