# OpenReview forum: "Self-Prompt: Leveraging Gradient-Based Search for Optimizing Prompts in Code Generation"
_ICLR.cc/2026/Conference — Submitted to ICLR 2026_

### Official Review · Reviewer_wcGw · 2025-10-26

**Soundness:** 2
**Presentation:** 2
**Contribution:** 2
**Rating:** 2
**Confidence:** 3

**Summary:**

This paper presents Self-Prompt, an automated prompt optimization framework designed for code generation tasks. Self-prompt based on the foundation of LLM self-sufficient prompt engineering, optimizing the prompt as a gradient-based search question in two phases. In the first phase, a base prompt generator generates a task-specific prompt, which is then filtered by heuristic rules to remove coding biases. In the second phase, self-prompt refines the prompt by identifying and selecting “code-sensitive” tokens through gradient analysis on the embedding layer. Experiments on three benchmarks across five LLMs demonstrate higher pass@1 than other baselines.

**Strengths:**

1. The Extra Prompt Generator Algorithm uses the gradient from the embedding layer to rank tokens for optimizing the prompt.

2. The experiment spanned five models and three datasets, covering different model architectures and scales.

**Weaknesses:**

1. The experiments only compare with APE, human, and general prompts. More recent and advanced prompt optimization methods are not discussed (e.g., PromptAgent).

2. The update rule(eq 5) lacks appropriate justification for the design of \alpha and update. Justification and alternative design are not discussed.

3. The gradient computation over embedding layers(Algorithm 2) introduces computational overheads, but efficiency analyses are not discussed.

4. The experiment only uses pass@1 as the evaluation metric, providing the result in pass@k would strengthen the paper and provide more details in a more robust generation.

5. RAG can also be viewed as a prompt refinement method, but the paper doesn't discuss

6. The comparison baselines are very weak.

7. The motivation is weak. Since it's a training-based method, directly comparing with training-free method is not fair. And authors didn't compare with finetuned models.

**Questions:**

1. Although the experiment demonstrate the effectiveness of self-prompt, concrete case study would help strengthen the paper.

2. How would the \alpha (eq. 5) design impact the final performance?

---

### Official Review · Reviewer_HWDw · 2025-10-28

**Soundness:** 2
**Presentation:** 3
**Contribution:** 1
**Rating:** 2
**Confidence:** 5

**Summary:**

This paper proposes Self-Prompt, an automated prompt optimization framework for code generation. The method consists of a base prompt generator and an extra prompt generator. It formulates prompt optimization as a search problem and refines the prompt via a gradient-based search. Experiments are conducted in five open-source LLMs across three benchmarks. Results show that Self-Prompt outperforms general prompts, human-crafted prompts, and APE in most settings.

**Strengths:**

1. The use of gradients and the token weight table to identify extra prompt tokens is an interesting and technically sound approach that leverages the model's internal state for prompt refinement.

2. The inclusion of ablation studies helps validate the contribution of each component.

**Weaknesses:**

1. The paper does not clearly specify the dataset used to train the extra prompt generator. It appears that the ground truth of the benchmark datasets is used for both prompt optimization (updating the weight table via loss computation) and evaluation.This leads to serious data leakage.

2. The baselines are relatively weak. Stronger prompt tuning or engineering methods (P-tuning, In-Context Learning etc.) should be included to demonstrate the effectiveness of Self-Prompt. Additionally, models in the main experiment are mostly smaller than 10B and were published over a year ago. It is unclear whether Self-Prompt generalizes well to recent larger base models or reasoning models.

3. Although the weight table plays a critical role in the extra prompt generator, there is little analysis of how the weight table and corresponding TOPK tokens evolve during training, or why these tokens contribute to performance gains. More interpretability experiments would help clarify the mechanism.

4. The weight table is updated using a heuristic rule based on loss differences. However, there is no theoretical or empirical justification provided for the convergence or stability of this update mechanism. It is unclear whether the process reliably improves prompts or if it can become stuck in suboptimal states.

5. There is a lack of running examples for Self-Prompt. The paper only provides a running example for APE, but not for Self-Prompt. This makes it difficult to understand how the proposed method works or how it overcomes the limitations of APE.

**Questions:**

1. Is the extra prompt a coherent sentence? Are the 50 tokens selected for the extra prompt semantically meaningful, or are they just a random assortment of tokens? Would replacing them with more coherent, domain-specific instructions or keywords result in better performance?

2. The problem formulation in Section 3.1 is unclear and not very intuitive. It seems that the actual optimization objective is to minimize cross-entropy loss over the vocabulary table and the ground truth. Could the authors clarify this and relate it more directly to the prompt optimization goal?

3. Heuristic filtering rules in base prompt generator. The paper mentions these rules, but no details are provided (L250). What are these rules, and how do they ensure that biased or low-quality prompts are removed?

---

### Official Review · Reviewer_Gz4X · 2025-10-31

**Soundness:** 2
**Presentation:** 2
**Contribution:** 3
**Rating:** 2
**Confidence:** 4

**Summary:**

The paper introduces self-promt, a prompt engineering technique that optimizes prompts for code generation with LLM. Self-prompt utilizes optimization to generate extra prompts using a finetuning process. Self-promt was able to outperform prior work APE in some cases on a selection of general-purpose small LLMs. The ablation study indicates that all components of self-prompt contribute to the overall performance of the proposed technique.

**Strengths:**

+ Improving code generation performance is important. Having better optimized prompts that cater to specific LLM capabilities potentially can give better results.

**Weaknesses:**

- The evaluation of the approach is not well justified and does not clearly indicate that the proposed technique is effective. The improvement is not consistent and sometimes similar to the original
- The choice of datasets is not optimal. HumanEval and MBPP are old datasets that are essentially solved by LLM.
- The choice of small LLMs is not justified. If foundation general-purpose LLMs have pretty much solved the benchmark (96%+ pass@1) using the original instruction, what is the purpose of evaluating self-promt on lesser models while extra training is required?
- The writing is hard to follow, lacking justification for various design choices. Some details are missing. It is not clear how the dataset was built to train the LLM for better prompt optimization. The evaluation section does not have good analysis text with a very small experiment result section, which hardly discusses anything useful and insightful.

**Questions:**

1. What is the loss function used to train the LLM? Does the technique require a dataset of prompts and corresponding generated code? If this is the case, what is the cost of generating this, and how is this dataset built with respect to the evaluation? Is this built on a portion of the evaluated dataset (HumanEval, MBPP, HumanEva+l, MBPP+)?
2. What is the justification for using the small open-source models to evaluate? On the HumanEval leader board, Qwen2.5 and DeepSeek-V3 are both doing well. Is it not better to use those to demonstrate that Self-prompt can improve even the best model? If not, in what scenario is Self-prompt applicable?
3. What are the reasons for inconsistent performance gain in some cases? Sof-prompt is not the best technique overall across the board.
4. What is the extra overhead from additional training? Would this extra overhead justify the gain, which is not always there?

---

### Official Review · Reviewer_bxCA · 2025-11-09

**Soundness:** 2
**Presentation:** 2
**Contribution:** 2
**Rating:** 2
**Confidence:** 4

**Summary:**

This paper introduces Self-Prompt, an automated prompt engineering framework that formulates prompt optimization for code generation as a search-based iterative refinement process using task-specific data.

**Strengths:**

1. The motivation of this paper is good.
2. The paper is well written and easy to follow.

**Weaknesses:**

1. The generalization of the paper’s conclusions is uncertain. Despite the availability of newer open-source models (e.g., Qwen3, Llama3), the experiments are conducted on older base models, which undermines the timeliness and applicability of the results.
2. Is it possible to report results with more k for pass@k evaluation to show the generalization of this work?
3. Missing citaion: the signal like reward from the model itself is a popular method in the community already (like self-rewarding[1]). While this paper does not discuss the related work in detail. The readers could not evaluate the novelty of this paper without enough discussion.
4. If I understand correctly, the proposed method does not involve complex training and thus should not require significant computational resources. What is the reason the paper does not include experiments or discussion on large-scale LLMs to validate the method’s scalability?

[1] Yuan, W., Pang, R. Y., Cho, K., Li, X., Sukhbaatar, S., Xu, J., & Weston, J. E. (2024, January). Self-rewarding language models. In Forty-first International Conference on Machine Learning.

**Questions:**

Refer to the weakness.

---

### Meta-Review · Area_Chair_H5iJ · 2026-01-08

**Summary:**

The paper proposes Self-Prompt, an automated and iterative prompt optimization framework for code generation that reframes prompt design as a search problem, and demonstrates across multiple benchmarks and open-source LLMs that it can match or outperform existing prompt engineering methods in terms of pass@k performance. The reviewers generally provide low scores and there is no author rebuttal submitted.

**Reviewer Concerns:**

The reviewers generally provide low scores and there is no author rebuttal submitted.

**Reviewer Scores:**

The reviewers generally provide low scores and there is no author rebuttal submitted.

---

### Decision · Program_Chairs · 2026-01-26

Reject